# Positive-Unlabeled Compression on the Cloud

**Yixing Xu**[†], **Yunhe Wang**[†], **Hanting Chen**[§], **Kai Han**[†],
**Chunjing Xu**[†], **Dacheng Tao**[‡], **Chang Xu**[‡]
[†]Huawei Noah's Ark Lab
[§]Key Laboratory of Machine Perception (MOE), CMIC,
School of EECS, Peking University, China
[‡]The University of Sydney, Darlington, NSW 2008, Australia
{yixing.xu, yunhe.wang, kai.han, xuchunjing}@huawei.com
htchen@pku.edu.cn, {dacheng.tao, c.xu}@sydney.edu.au

## Abstract

Many attempts have been done to extend the great success of convolutional neural networks (CNNs) achieved on high-end GPU servers to portable devices such as smart phones. Providing compression and acceleration service of deep learning models on the cloud is therefore of significance and is attractive for end users. However, existing network compression and acceleration approaches usually fine-tuning the svelte model by requesting the entire original training data (*e.g.* ImageNet), which could be more cumbersome than the network itself and cannot be easily uploaded to the cloud. In this paper, we present a novel positive-unlabeled (PU) setting for addressing this problem. In practice, only a small portion of the original training set is required as positive examples and more useful training examples can be obtained from the massive unlabeled data on the cloud through a PU classifier with an attention based multi-scale feature extractor. We further introduce a robust knowledge distillation (RKD) scheme to deal with the class imbalance problem of these newly augmented training examples. The superiority of the proposed method is verified through experiments conducted on the benchmark models and datasets. We can use only $8\%$ of uniformly selected data from the ImageNet to obtain an efficient model with comparable performance to the baseline ResNet-34.

## 1 Introduction

Convolutional neural networks (CNNs) have been widely used in a variety of computer vision applications such as image classification [14, 18, 21], object detection [5], semantic segmentation [17], clustering [31], multi-label learning [23], *etc*CNNs are often over-parameterized to achieve a good recognition performance. However, many empirical studies suggest that those redundant parameters or filters can be eliminated without affecting the performance of the network. To be compatible with various running environments (*e.g.* cell phone and autonomous driving) in real-world applications, well trained neural networks need to be further compressed and accelerated accordingly. Considering the scalable computation resource (*e.g.* GPU and RAM) offered by the cloud, it is therefore promising to provide network compression service for end users.

Compared with the model compression service offered by the cloud, it would be much harder for end users to compress the cumbersome network by themselves. One one hand, GPUs are essential to doing effective deep learning. Compared with setting up their own servers, many users tend to spin up cloud instances with GPUs by balancing the flexibility and the investment, especially when the GPUs are only needed for several hours. One the other hand, not every user is a deep learning expert, and a cloud service would be expected to produce efficient deep neural networks according to users' needs.

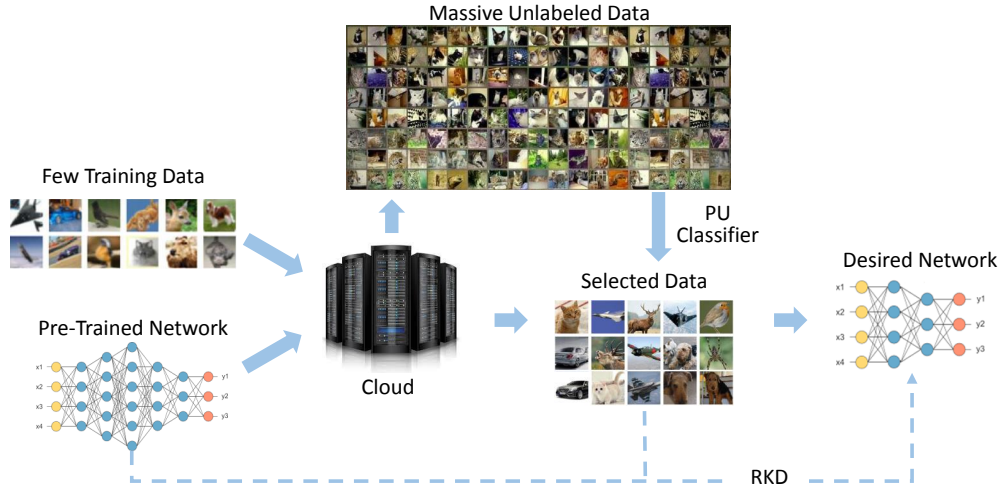

Figure 1: The diagram of the proposed method for compressing deep networks on the cloud.

Existing methods like quantization approach [6], pruning approach [4] and knowledge distillation approach [9] cannot be easily deployed on the cloud to compress the cumbersome network submitted by end customers. The major reason is that most of these methods require users to provide the original training data for fine-tuning the compressed network to avoid much drop of the accuracy. However, compared with the model size of modern CNNs, the size of the entire training data would be much larger. For example, ResNet-50 [8] only occupies an about $95MB$ for storing its parameters while its training dataset (*i.e.* ImageNet [14]) contains more than one million images with an over $120GB$ file size. Therefore, given the limitation of transmission speed (*e.g.* $10MB$/s), users have to wait for a long period of time before launching the compression methods, which does harm to user experience of the service.

In this paper, we suggest a two-stage pipeline to leverage the easily accessible unlabeled data for training compact neural networks, as shown in Fig. 1. Users are required to upload the pre-trained deep network and a small portion (*e.g.* $10\%$) of the original training data. Taking the scarce labeled data as 'positive', in the unlabeled pool (*e.g.* Flickr [12]) there could be 'positive' data that follows a similar distribution (*e.g.* of the same concept), while the remaining data are treated as 'negative'. A binary PU classifier learned from these positive-unlabeled data can then be employed to identify the most related unlabeled data to augment the training set for our compression task. In order to correct the biased labels contained in the augmented dataset, we further develop a robust knowledge distiller (RKD) to address the problem of noisy and imbalanced labels. Experimental results conducted on several benchmark datasets and deep models demonstrate that with the help of massive unlabeled data, the proposed method is effective for learning efficient networks with only a small proportion of the original training data.

## 2   Positive-Unlabeled Classifier for More Data

Here we first present some preliminaries for learning efficient neural networks, and then develop a novel framework to effectively utilize massive unlabeled data on the cloud for training.

### 2.1   Knowledge for Compressing Neural Networks

Conventional deep model compression algorithms aim to eliminate redundant weights or filters in pre-trained deep neural networks. The resulting networks are often of specific structures such as sparse matrices and mix-bit multiplications which need additional technical supports. In contrast, knowledge distillation (KD) method [9] is proposed to directly learn student networks with fewer parameters and computational complexities by inheriting feature information from the given teacher network.

Denoting the pre-trained teacher network and the desired efficient student networks as $\mathcal{N}_{te}$ and $\mathcal{N}_{st}$, respectively, the student network is trained using the following objective function:

$$\mathcal{L}_{KD} = \frac{1}{n} \sum_i \mathcal{L}_c(\mathbf{y}_i^{te}, \mathbf{y}_i^{st}),\tag{1}$$

where $\mathcal{L}_c(\cdot, \cdot)$ is the cross-entropy loss, $n$ is the number of samples in the training set, $\mathbf{y}_i^{te}$ and $\mathbf{y}_i^{st}$ are the output responses corresponding to the teacher network $\mathcal{N}_{te}$ and student network $\mathcal{N}_{st}$, of the same input data $\mathbf{x}_i$, respectively. By using the KD method described in Eq. 1, the student network is able to generalize in the same way as the teacher network, and can empirically obtain a much better result than training it from-scratch.

However, the number of samples in the training dataset of $\mathcal{N}_{te}$ is often extremely large, *e.g.* there are over 1.2 million images in the ILSVRC 2012 dataset with file size of 120*GB*. Differently, modern CNN architectures are more and more lightweighted, *e.g.* the model size of MobileNet-v2 [22] is only about 15*MB*. Thus, the time consumption of uploading such huge datasets affects the user's experience on the model compression service on the cloud.

## 2.2 Positive-Unlabeled Classifier for Selecting Data

In order to reduce the required number of samples $n$ in the training dataset, we propose to look for alternative data. Actually, there are massive datasets on the cloud servers for conducting different tasks (*e.g.* CIFAR, ImageNet and Flickr). We can regard them as unlabeled data, and a small proportion of the original samples as positive data. Thus, the data selection task is exactly a positive-unlabeled (PU) learning problem [13, 30].

PU learning method focuses on learning a classifier from positive and unlabeled data. Given $\mathbf{x}_i \in \mathcal{X} \subseteq \mathbb{R}^d$, $y_i \in \mathcal{Y} = \{\pm 1\}$ be the input samples and output labels, together with $n_l$ and $n_u$ be the number of labeled and unlabeled samples, respectively. The training set $T$ of the PU classifier can be formulated as:

$$T = L \cup U = \{(\mathbf{x}_l, +1)\}_{l=1}^{n_l} \cup \{(\mathbf{x}_u, y_u)\}_{u=1}^{n_u},\tag{2}$$

where $L$ is the labeled set and $U$ is the unlabeled set, respectively.

Denote the desired decision function as $f : \mathcal{X} \to \mathcal{Y}$, and $F : \mathcal{X} \to \mathbb{R}$ is a discriminant function that maps the input data to a real number such that $f(\mathbf{x}) = sign(F(\mathbf{x}))$, and $l : \mathcal{X} \times \mathcal{Y} \to \mathbb{R}$ is an arbitrary loss function, the decision function $f$ can be optimized by the following equation:

$$\tilde{R}_{pu}(f) = \pi_p \hat{R}_p^+(f) + \max\{0, \hat{R}_{\mathbf{x}}(f) - \pi_p \hat{R}_p^-(f)\},\tag{3}$$

where $\hat{R}_{\mathbf{x}}(f) = \mathbb{E}_{\mathbf{x}}[l(F(\mathbf{x}), -1)]$, $\hat{R}_p^+(f) = \mathbb{E}_p[l(F(\mathbf{x}), +1)]$ and $\hat{R}_p^-(f) = \mathbb{E}_p[l(F(\mathbf{x}), -1)]$ are the corresponding risk functions, and $\pi_p = p(y = +1)$ is the class prior. $l(t, y)$ is an arbitrary loss function between the target $t$ and the ground truth label $y$.

For the given pre-trained teacher network $\mathcal{N}_{te}$, we can ask the user to provide a tiny dataset $L^t$ consists of a small proportion (*e.g.* 10%) of the original training set. Then we can collect an unlabeled dataset $U$ on the cloud, Eq. 3 can be further utilized to select more positive data from $U$ to construct another training dataset for conducting the subsequent model compression task.

Since the pre-trained teacher network $\mathcal{N}_{te}$ is designed to solve the original tasks, such as an ordinary classification, it is infeasible to directly use the same architecture on PU classification [29]. Therefore, we introduce an attention based multi-scale feature extractor $\mathcal{N}_A$ for extracting features of input data, *i.e.* $F(\mathbf{x})$. Note that the deep features transition from general to specific along the network, and the transferability of the features drop rapidly in higher layers. Simply using the feature produced by the last layer will produce a large transferability gap, while using combined features from layers in different locations of the network will reduce the gap.

Specifically, let $\mathbf{h}^j \in \mathbb{R}^{H^j \times W^j \times C^j}$ be the features extracted in the $j$-th layer. Note that these outputs cannot be directly concatenated because the size of heights and widths are different. A common way to mitigate this problem is using global average pooling. Given $\mathbf{h}^j$, the global spatial information is compressed into a channel-wise descriptor $\mathbf{o}^j \in \mathbb{R}^{C^j}$ [10], where the $c$-th element of $\mathbf{o}^j$ is calculated by:

$$o_c^j = \frac{1}{H \times W} \sum_{m=1}^{H} \sum_{n=1}^{W} h(m, n, c).\tag{4}$$

Given the compressed channel-wise descriptors, a simplest way is to directly concatenate them together into a single vector. However, it is not flexible enough for the vector to reflect the importance of the input signals, which represent features from general to specific. Generally, inputs containing more information should have a larger weight. Thus, we add attention on top of these descriptors for adaptation between modalities. Attention method can be viewed as a way to allocate the input signal so that more informative component will get more attention by the next layer, which has been widely used in CNN across a range of tasks [1, 11]. Specifically, given the concatenated channel wise descriptor $\mathbf{o} = \{\mathbf{o}^1, \cdots, \mathbf{o}^j\}$, we opt to employ a gating mechanism as suggested in [10]:

$$\mathbf{w} = Attention(\mathbf{o}, \mathbf{W}) = \sigma(\mathbf{W}_2 \delta(\mathbf{W}_1 \mathbf{o})), \quad (5)$$

---

**Algorithm 1** PU classifier for more data.

**Require:** An initialized network $\mathcal{N}_A$, a tiny labeled dataset $L^t$ and an unlabeled dataset $U$.
1: **Module 1: Train PU Classifier**
2: **repeat**
3:    Randomly select a batch $\left\{\mathbf{x}_i^U\right\}_{i=1}^N$ from $U$ and $\left\{\mathbf{x}_i^{L^t}\right\}_{i=1}^N$ from $L^t$;
4:    Optimize $\mathcal{N}_A$ following Eq. 3;
5: **until** convergence
6: **Module 2: Extend the labeled dataset**
7: Obtain the positive data $U^p$ from $U$ utilizing PU classifier $\mathcal{N}_A$;
8: Unify the positive dataset $U^p$ and tiny dataset $L^l$ to achieve extended dataset $L^l = L^t \cup U^p$;

**Ensure:** Extended dataset $L^l$.

---

in which $\delta$ is the ReLU transformation, $\mathbf{W}_1 \in \mathbb{R}^{\frac{C}{r} \times C}$ and $\mathbf{W}_2 \in \mathbb{R}^{C \times \frac{C}{r}}$ are the parameters of two FC layers that reduce the dimensionality of the input by a ratio $r$, followed by a non-linearity and then increase the dimensionality back to origin. A sigmoid activation is used to perform the attention weight $\mathbf{w}$. The final output $\tilde{\mathbf{o}}^j$ is obtained by simply re-scaling the channel wise descriptor:

$$\tilde{\mathbf{o}}^j = w_j \mathbf{o}^j. \quad (6)$$

Based on the proposed feature extractor $\mathcal{N}_A$, we train the data from the unlabeled dataset $U$ and the tiny labeled dataset $L^t$ which is randomly sampled from the original dataset $L$. Dataset $L^t$ is then expanded with the data which is classified as positive in dataset $U$, and finally derive a larger positive dataset $L^l$ with non-negative PU loss Eq. 3. Specifically, we minimize the non-negative PU loss with stochastic gradient descent (SGD) and stochastic gradient ascent (SGA). Denoting $t_i = \hat{R}_{\mathbf{x}}(f(\mathbf{x}_i^U)) - \pi_p \hat{R}_p^-(f(\mathbf{x}_i^{L^t}))$. When $t_i > 0$, we minimize Eq. 3 with SGD. Otherwise, the gradient of $t_i$ is computed and we update the parameter of the network with SGA. That is, we go along with $- \bigtriangledown_\theta t_i$, in order to alleviate the over-fitting of the current mini-batch $i$. A more specific procedure is presented in Algorithm 1.

## 3 Robust Knowledge Distillation

The number of training examples in each class is usually balanced for a better training of deep neural networks. However, the dataset $L^l$ generated by PU learning may suffer from data imbalanced problem. For example, in ImageNet dataset the number of samples in category 'dog' is 30 times more than that in category 'plane', and there are no sample from category 'deer'. When $L^t$ is randomly sampled from CIFAR-10 and ImageNet is treated as the unlabeled dataset $U$, the number of 'dog' samples will dominate the expanded dataset $L^l$. Therefore, it is unsuitable to directly adopt the KD method given the imbalanced dataset $L^l$.

There are many works which focused on the data imbalanced problem. However, they cannot be directly used in our problem, since the number of samples in each category is unknown in $L^l$. The PU learning method only distinguish whether the images in $U$ belong to the given dataset $L$, but never deal with the specific classes of input images.

In practice, we utilize the output of the teacher network. Note that instead of treating the output class label as the ground truth of the input sample $\mathbf{x}_i$, we treat the output response $\mathbf{y}_i^{te} = softmax(Z_i/T)$ as the pseudo ground truth vector, in which $Z_i$ is the final score output and $T$ is the temperature parameter which helps soften the output when the probability for one class is close to 1 and others are close to 0 ($T = 1$ in the following experiments). To this end, we propose a robust knowledge distillation (RKD) method to solve the data imbalanced problem.

Specifically, we assign weight to each category of the samples, where categories with fewer samples will have larger weights. Based on this principle, defining $\mathbf{y} = \{y^1, y^2, \cdots, y^K\} = \sum_i \mathbf{y}_i^{te}$, we have

the weight vector $\mathbf{w}_{kd} = \{w_{kd}^k\}_{k=1}^K$, in which:

$$w_{kd}^k = \frac{K/y^k}{\sum_{k=1}^K 1/y^k}, \quad k = 1, 2, \cdots, K, \tag{7}$$

and $K$ is the number of categories in the original dataset. When training the student network, the weight of the input sample is defined as $w_i = w_{kd}^{category_i}$ in which $category_i$ is the index of the largest element in the ground truth vector $\mathbf{y}_i^{te}$.

Therefore, the surrogate KD loss can be derived based on Eq. 1:

$$\tilde{\mathcal{L}}^{KD} = \frac{1}{n} \sum_i w_i \mathcal{F}_{ce}(\mathbf{y}_i^{te}, \mathbf{y}_i^{st}). \tag{8}$$

Note that the derivation of $\mathbf{w}_{kd}$ is not optimal, since the predicted output response $\mathbf{y}_i^{te}$ is not optimal and is contaminated with noise. However, we assume that the teacher network is well-trained, and there is only a slight difference between the elements in $\mathbf{w}_{kd}$ and the optimal weight vector $\mathbf{w}_{kd}^*$:

$$p(|w_{kd}^k - w_{kd}^{*k}| < \epsilon) > 1 - \delta. \tag{9}$$

Thus, we give a random perturb $\epsilon$ on each element of the original weight vector $\mathbf{w}_{kd}$ and get a finite set of possible weight vectors $\mathbf{W} = \{\mathbf{w}_{kd\_1}, \cdots, \mathbf{w}_{kd\_n}\}$, in which $|w_{kd\_i}^k - w_{kd}^k| < \epsilon$. Note that this is similar to the cost-sensitive learning with multiple cost matrices. Based on these weight vectors, we are able to train the student network with the following equation:

---

**Algorithm 2** Robust Knowledge distillation.

**Require:** A given teacher network $\mathcal{N}_{te}$, the extended dataset $L^l$ and a hyper-parameter $\epsilon$.
1: Initialize the student network $\mathcal{N}_{st}$;
2: Calculate weight vectors $\mathbf{w}_{kd}$ using Eq. 7 and generative a set $\mathbf{W}$ using a random perturb $\epsilon$;
3: **repeat**
4:     Randomly select a batch $\left\{ \mathbf{x}_i^{L^l} \right\}_{i=1}^m$;
5:     Employ the teacher and student network: $\mathbf{y}_i^{te} \leftarrow \mathcal{N}_{te}(\mathbf{x}_i^{L^l}); \mathbf{y}_i^{st} \leftarrow \mathcal{N}_{st}(\mathbf{x}_i^{L^l})$
6:     Calculate the surrogate KD loss $\tilde{\mathcal{L}}^{KD}$ following Eq. 8;
7:     Update $\mathcal{N}_{st}^{\mathbf{W}}$ with Eq. 10;
8: **until** convergence
**Ensure:** The student network $\mathcal{N}_{st}$.

---

$$\mathcal{N}_{st}^{\mathbf{W}} = \arg \min_{\mathcal{N}_{st} \in \mathcal{N}} \max_{\mathbf{w} \in \mathbf{W}} \tilde{\mathcal{L}}^{KD}(\mathcal{N}_{st}, \mathbf{w}), \tag{10}$$

in which $\mathcal{N}$ is the hypothesis space. This is similar to the method proposed in [27]. However, different from the cost matrix, the weight vector in Eq. 7 is only related to the proportion of the samples in each category and has nothing to do with the classification result, which is suitable for our learning problem. Besides, we solve a multi-class problem rather than a binary class problem.

## 4 Experiments

### 4.1 CIFAR-10

The widely used CIFAR-10 benchmark is first selected as the original dataset, which is composed of $32 \times 32$ images from 10 categories. We randomly select $n_l$ samples in each class and form the tiny labeled dataset $L^t$ with $10n_l$ positive samples. Benchmark dataset ImageNet contains over $1.2M$ images from 1000 classes, but it is treated as the unlabeled dataset $U$ with $n_u = 1.2M$ unlabeled samples in our experiment. In this setting, 'positive' indicates that the category of the input sample belongs to one of the categories of the original dataset CIFAR-10. Recall that the class prior $\pi_p = p(y = +1)$ in Eq. 3 indicates the proportion of the positive samples in $U$, which is assumed to be known in the following experiments. In practice, it can be estimated with the method in [19]. In this experiment, we manually select positive data from $U$ based on the name of the category provided by ImageNet 2012 classification dataset [14], and train the student network with manually selected data using the proposed RKD method as the baseline. The total number of positive data we selected is around $270k$, thus we set the class prior $\pi_p = 0.21 \approx 270k/1280k$ in the following experiment.

The model used in the first step is an attention based multi-scale feature extractor based on ResNet-34. Specifically, the channel-wise descriptor $\mathbf{o}^j$ in Eq. 4 is derived from the outputs of 4 groups in ResNet-34. The network is trained for 200 epochs using SGD. We use a weight decay of 0.005 and

Table 1: Classification results on CIFAR-10 dataset. The best results are bold in the table.

| Method | $n_l$ | $n_t$ | Data source | FLOPs | #params | Acc(%) |
|--------|-------|-------|-------------|-------|---------|--------|
| Teacher | - | 50,000 | Original Data | 1.16G | 21M | 95.61 |
| KD [9] | - | 50,000 | Original Data | 557M | 11M | 94.40 |
| Baseline-1 | - | 269,427 | Manually selected data | 557M | 11M | 93.44 |
| Baseline-2 | - | 50,000 | Randomly selected data | 557M | 11M | 87.02 |
| PU-s1 | 100 | 110,608 | PU data | 557M | 11M | **93.75** |
| | 50 | 94,803 | | | | 93.02 |
| | 20 | 74,663 | | | | 92.23 |
| PU-s2 | 100 | 50,000 | PU data | 557M | 11M | 91.56 |
| | 50 | 50,000 | | | | 91.33 |
| | 20 | 50,000 | | | | 91.27 |

momentum of 0.9. We start with a learning rate of 0.001 and divide it by 10 every 50 epochs. Data in ImageNet is resized to $32 \times 32$ rather than $224 \times 224$ in our experiment. Random flipping, random crop and zero-padding are used for data augmentation. In the second step, the teacher network is a pre-trained ResNet-34, and ResNet-18 is used as the student network. A weight decay of 0.0005 and momentum of 0.9 is used. We optimized the student network using SGD by starting with a learning rate of 0.1 and divide it by 10 every 50 epochs. $\pi_p = 0.21$ is used in the following experiments.

Note that in the first step in our algorithm, the positive samples are automatically selected by the PU method. Thus, the number of training samples for the second step is unfixed, and could be influenced by the architecture of the network, the hyper-parameter used in the experiment, etc. In this circumstances, it is difficult to judge whether a good result is benefit from the quality or the number of the training data. Therefore, there are two settings in our experiment. The first setting is to feed all the positive data selected by the PU method to the second step to train the student network. Another setting is to randomly select a bunch of data which has the same number as the original training dataset ($50k$ for CIFAR-10).

The experimental results are shown in Tab. 1. Wherein, 'Baseline-1' method directly feeding manually selected positive data to the second step. 'Baseline-2' method randomly select 50000 data and then fed to the second step, which inevitably contains many negative data and should results in a bad performance. 'PU-s1' is the setting of feeding all the positive data selected by the PU method to the second step, and 'PU-s2' is the setting of randomly feeding 50000 positive data to conduct the second step. In addition, $n_l$ is the number of samples selected from each class in CIFAR-10, $n_t$ is the number of training samples used to train the student network. Suppose that $n_p^u$ positive samples are selected from $U$ by PU method, then we have $n_t = n_p^u + 10n_l$.

The result shows that the performance of the proposed method is even better than the baseline method. With 1000 samples in CIFAR-10 and about $110k$ training samples selected from ImageNet, it achieves a higher accuracy than the baseline method with $270k$ manually selected training data. It shows the priority of the proposed method of selecting high quality positive samples from unlabeled dataset. In fact, manually selecting positive samples from ImageNet requires a huge effort, and the way we select are not carefully enough to exclude all the negative data in the manually selected dataset.

In the previous experiments the class prior $\pi_p$ is assumed to be known. In practice we may suffer from the error of estimating $\pi_p$. Thus, a number of different $\pi_p'$ are given to the proposed algorithm in order to test the robustness of the proposed method on the class prior. All the experimental settings are exactly the same except for the change from $\pi_p$ to $\pi_p'$. Fig. 2 shows the classification accuracies of using different $\pi_p'$. $50k$ training samples are randomly selected in the second step to alleviate the influence of the number of training samples. The same experiments are conducted on both ResNet-34 and the attention based multi-scale feature extractor with traditional KD and RKD method to show the superiority of the proposed architecture and RKD method. The result shows that the proposed architecture with RKD method behaves the best, and is more robust on the under-estimate and over-estimate of the true class prior $\pi_p$.

The experimental results show that although there are many negative data in the Imagenet dataset, the PU classifier can successfully pick a large amount of positive data whose categories is the same as that of given data. Therefore, the extended dataset with given data and selective data can be used to train a portable student network.

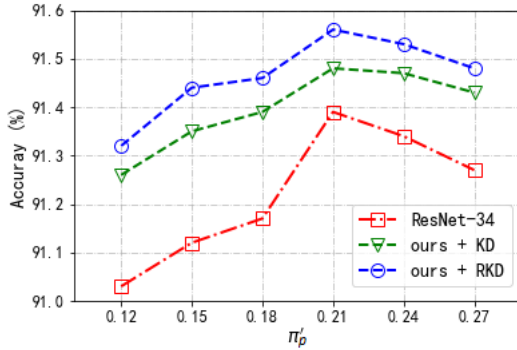 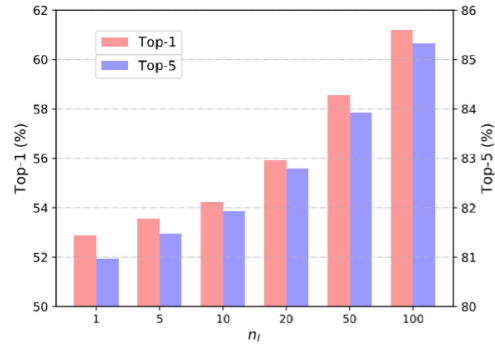

Figure 2: Classification accuracies on CIFAR-10 dataset with different $\pi'_p$.

Figure 3: Relationship between the number of samples selected from each category in ImageNet and the resulting accuracy.

Table 2: Classification results on ImageNet dataset. "KD-all" utilizes the entire ImageNet training dataset to train the student network. "KD-500k" randomly selects $500k$ training data from ImageNet for learning the student network.

| Algorithm | $n_t$ | Data source | FLOPs | #params | top-1 acc(%) | top-5 acc(%) |
|---|---|---|---|---|---|---|
| Teacher | 1,281,167 | Original Data | 3.67G | 22M | 73.27 | 91.26 |
| KD-all | 1,281,167 | Original Data | 1.82G | 12M | 68.67 | 88.76 |
| KD-500k | 500,000 | Original Data | 1.82G | 12M | 63.90 | 85.88 |
| PU-s1 | 690,978 | PU data | 1.82G | 12M | 61.92 | 86.00 |
| PU-s2 | 500,000 | PU data | 1.82G | 12M | 61.21 | 85.33 |

## 4.2 ImageNet

Then, we conduct experiment on ImageNet dataset, which is treated as the original dataset. Flicker1M dataset is used as the unlabeled dataset[1]. The experimental setting is the same as those in the CIFAR-10 experiments, except that we train 110 epochs in both steps and divide the learning rate by 10 every 30 epochs. The class prior $\pi_p$ is set to 0.7 in the following experiments. Experimental result is shown in Tabel 2.

In order to make a fair comparison, we randomly select $500k$ samples from ImageNet and treat KD-500k as the baseline method. In the proposed method, we randomly select 100 samples from each category in ImageNet and form a tiny labeled dataset $L^t$, and then PU method is used to select positive data from Flicker1M dataset. The result shows that when feeding all the positive samples to the second step, the top-5 accuracy is even better than the baseline method. The reason that top-1 accuracy is worse than the baseline while top-5 is better is that we do not distinguish the specific category when using the PU method. Thus, the proposed method is better at learning meta knowledge than the specific label. When using a same number of training samples, the proposed method has only 0.5% top-5 accuracy drop compared to the baseline method while using only 8% of the samples in the original dataset.

Fig. 3 shows the relationship between the number of samples selected from each category in ImageNet and the accuracy of the proposed method. It is obvious that our method still achieves a promising result when using only about 0.8% samples of the original dataset.

## 4.3 MNIST

Since most of experiments in existing methods are conducted on the MNIST dataset, we further conduct the experiments on this dataset in order to compare our method to the state-of-the-art methods

Table 3: Comparsion on the state-of-the-art methods on the MNIST dataset.

| | 1 | 2 | 5 | 10 | 20 | all-meta-data |
|---|---|---|---|---|---|---|
| data-free KD [16] | - | - | - | - | - | 92.5 |
| FitNet [20] | 90.3 | 94.2 | 96.1 | 96.7 | 97.3 | - |
| FSKD [15] | 95.5 | 97.2 | 97.6 | 98.0 | 98.1 | - |
| PU-s1 | **98.5** | **98.7** | **98.7** | **98.8** | **98.9** | - |
| PU-s2 | 98.3 | 98.5 | 98.5 | 98.6 | 98.6 | - |

including FitNet [20], FSKD [15] and data-free KD method [16]. The EMNIST dataset[2] is used as the unlabeled dataset, which contains $814K$ hand-written letters and digits. We randomly select 1,2,5,10 and 20 samples from each category in MNIST to form the tiny set $L^t$. We use a standard LeNet-5 as the teacher network and the student network is 'half-size' to that of the corresponding teacher network in terms of the number of feature map channels per conv-layers. The class prior $\pi_p$ is set to $0.47$ in the following experiments.

Detailed classification results are shown in Tab. 3. It is clear that the proposed method outperforms FitNet and FSKD with a notable margin and is more robust when the number of labeled samples in each category is extremely rare ($< 5$).

## 5 Related Works

In this section, we give a brief introduction about the related works of model compression.

There is a bunch of algorithms designed for learning efficient neural networks with fewer memory usage and computational complexity [7, 28]. For example, Gong *et.al.* [6] investigated the vector quantization approach for representing similar weights for smaller CNNs. Denton *et.al.* [4] exploited the redundancy within convolutional filters to derive approximations and significantly reduced the required computational costs. Chen *et.al.* [3] compressed the weights in neural networks using the hashing trick [24, 25]. Hinton *et.al.* [9] presents the knowledge distillation approach for transferring information from the pre-trained teacher network to a compressed student network.

Nowadays, there are only a few attempts to learn efficient neural networks with some meta-data of the training set or without using the original training data. For instance, Srinivas and Babu [26] directly removed the redundant similar neurons in a systematic way. Based on knowledge distillation, Lopes *et.al.* [16] used some extra meta-data to learning smaller deep neural networks. However, the performance of the resulting networks learned through these methods are often much worse than that of the baseline network. This is because the amount of available data and information is extremely small. More recently, Chen *et.al.* [2] designed a generator for generating data of the similar properties as those of the original dataset, which obtained promising performance but lacked efficiency for generating images.

## 6 Conclusion

Most of existing network compression methods require the original dataset to achieve acceptable performance. However, the huge size of the training dataset leads to unacceptable transmission cost from end-user to the cloud. Therefore, we propose a two-step framework to compress the given neural network using only a small portion of the training data. Firstly, a PU classifier with an attention based multi-scale feature extractor is trained with the given labeled data and massive unlabeled data on the cloud. Then, a new dataset is conducted by combining the given data and the 'positive' data selected by PU classifier. Secondly, we develop a robust knowledge distillation (RKD) method to address the class imbalanced problem with noise in the augmented dataset. Experiments on the MNIST, CIFAR-10 and ImageNet datasets demonstrate that the proposed method can successfully dig more useful training samples using only a small amount of original data, and achieve the state-of-the-art performance comparing to other few-shot learning model-compression methods.

**Acknowledgments**

We thank anonymous area chair and reviewers for their helpful comments. Chang Xu was supported by the Australian Research Council under Project DE180101438.

## Footnotes

[1]http://press.liacs.nl/mirflickr/mirdownload.html

[2]https://www.westernsydney.edu.au/bens/home/reproducible_research/emnist

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
