[Reviews · NeurIPS 2019]

Reviewer 1



The paper targets the application of network compression using a cloud platform. Instead of uploading all the training data onto the platform, the paper suggests uploading a small portion of data as positive (P) data and use larger datasets already on the platform as unlabeled (U) data. After training a PU classifier, the classifier will be used to select more P data from the U data. And such selected data, together with the original data, are used in a knowledge distillation framework to compress the original network. The experimental results show that the compressed network’s performance is close to the original deep neural network trained on all data, on three widely used datasets. Originality, The paper is a smart application of PU learning. The idea of using PU learning to do data augmentation is smart and does make sense. However, the application problem of network compression on the cloud may be problematic. The users have both the well-trained network, as well as the data to train the network. In this way, compressing the network is easier on the users’ part. If a cloud platform is used, then the whole big network can be executed on the cloud, and thus no compression is required. In this way, the application does not make sense. Other applications that can motivate the paper should be searched. For the technology part, the paper proposes some heuristic of feature extraction and knowledge distillation, by slightly modifying the original attention and knowledge distillation methods. These modifications are targeted at the current problem, but the novelty is limited. Quality, Although the compressed network can comparable performance, why the paper works are still not clear. More comparisons are required: For example, the proposed method may work not only due to the “PU” learning part, but also the attention-based feature extraction part. In this way, is there any comparison on using the attention-based feature extraction and not use it? Another thing is, the proposed method may also work due to the robust knowledge distillation method. In this way, another possible comparison is that using a part of, or all the unlabeled data set with knowledge distillation to compress the network. With this comparison, it will be clear what is the contribution of each component of the proposed method. Clarity, The paper is clearly written. There are two minor problems. One is that referring to an equation is usually written as Eq.(x) instead of Fcn.(x). Another is some of the references that are not actually cited in the text, for example, [12,18,21,23,25,27,28]. Significance, The paper is a nice try of PU learning on data augmentation application. However, its application problem may not be appropriate enough in practice. Some necessary comparisons are also missing. ------------------------------------------------ I am satisfied after reading the rebuttal and would like to increase my score. However, the application motivation should be strengthened in a revised version.

Reviewer 2



Generally speaking, this is a high quality paper which solve the compression problem in a different but effective way. By utilizing the proposed algorithms, the end users do not need to spend hours uploading dataset to the cloud, which is quite friendly and attractive for end users. The paper leverages the strength of PU learning in augmenting data and the strength of KD method in compression. The experimental results show that the network can be compressed effectively with only 8% of the original data. The paper is well-written and easy to understand.

Reviewer 3



This paper focuses on solving model compression problem with limited labeled data. Taking a pre-prepared large-scale dataset as unlabeled, a handful of data from the original training data as positive, it is an interesting way to select data for the subsequent compression task with the help of a PU classifier. Robust KD method is used to deal with the noise and the data imbalanced problem. Multi-feature network with attention structure alleviate the dimensionality gap between datasets. This two-step method is clear and efficient, and the experimental results are very impressive. ----------------------- I am satisfied with the rebuttal.

[Author Response · NeurIPS 2019]

We sincerely thank the three reviewers for their constructive comments and supports.

**Response to Reviewer #1:**

**Q1:** Compressing on the user part. **A:** GPUs are essential to doing effective deep learning. Compared with setting up
their own servers, many users tend to spin up cloud instances with GPUs by balancing the flexibility and the investment,
especially when the GPUs are only needed for several hours. In addition, not every user is a deep learning expert, and
thus a cloud service would be expected to produce efficient deep neural networks according to users' needs.

**Q2:** Cloud platform is used, no compression is required. **A:** The compressed networks are often deployed on low-end
computing devices, e.g. digital cameras and mobile phones. By doing this, the cloud service latency can be avoided and
the application can be well executed even there is no Internet connection, which will improve the user experience.

**Q3:** Novelty. **A:** The novelty of this paper is twofold. Firstly, compressing network with little labeled data is a
challenging task that has rarely been investigated by existing works in the field. To compensate the lack of labeled data,
we introduce PU learning to select the most related data from a large pool of unlabeled data for the compression task.
Secondly, we enhance the robustness of knowledge distillation to deal with data imbalance problem and noise. An
attention based multi-scaled feature extractor is developed to cope with PU data better.

**Q4:** Comparison on using the attention-based feature extraction or not. **A:** The results are shown in Figure 2. The blue
line is the result on using attention-based feature extractor. The red line is the result without attention-based feature
extractor. The superiority of the proposed architecture leads to the accuracy improvement from $91.39\%$ to $91.57\%$.

**Q5:** Comparison on part of, or all the unlabeled data set with KD method. Using robust KD or not. **A:** We include new
experiments on CIFAR-10. Randomly choosing 50,000 samples leads to a $87.02\%$ accuracy ($91.56\%$ for PU method
choosing 50,000 samples). The result is bad since many negative data are selected in this way and is used to train the
student network. Using all $1.2M$ unlabeled data leads to a $94.01\%$ accuracy since all the positive data are guaranteed to
be selected, but is about 12 times slower than using PU data ($93.75\%$ for PU method choosing $0.1M$ samples). Robust
KD leads to $0.6\%$ increase.

**Q6:** Minor problems. **A:** Thanks for this nice concern. All these typos will be corrected in the final version.

**Response to Reviewer #2:**

**Q1:** Figure 4. **A:** The first row represents the data in the original dataset. The second is the positive data selected by PU
classifier. The third is the negative data selected by PU classifier. Spider is in both selected data and negative data, since
the PU classifier does not classify unlabeled data with $100\%$ accuracy, and spider represents the noise in positive data.

**Response to Reviewer #3:**

**Q1:** Advantages over related works. **A:** Theoretically, we utilize the strength of PU learning in augmenting data, and
the strength of KD method in compressing neural networks, which is suitable for solving compression problem with
little labeled data. Besides, we propose multi-feature network with attention and robust KD method to better solve
the problem. Experimentally, we compare with the state-of-the-art methods in Table 3, and the accuracy results show
the priority of the proposed method. Generally speaking, the proposed method is robust on the number of positive
samples, and performs better. In other methods, the student network is compressed by pruning [1] or re-normalization
[2] the giant teacher network with unlabeled data, which means that the detail architecture of the teacher network is
required. In the proposed method, we only need the input and output interface of the teacher network instead of the
whole architecture, which is more flexible for users to protect their own privacy.

**Q2:** The motivation to tackle imbalanced data problem is unclear. **A:** PU classifier treats all samples in the training set
as 'positive', the specific category is undistinguishable. PU may select lots of data for some positive categories, while
little for others. This causes data imbalanced problem. Eq.(7) and (8) are used to tackle the imbalanced data problem.
We utilize the output of teacher network, and assign larger weights to categories with fewer samples. The weighted
surrogate KD loss (Eq.(8)) can alleviate the imbalanced data problem.

**Q3:** Class prior $\pi_p$. **A:** In CIFAR-10 experiments, it is set equal to the ratio of manually selected data. In MNIST
experiments, it is set to the real ratio of numbers in EMNIST. In ImageNet experiments, which is much like the reality
settings, $\pi_p$ is estimated by the prior estimation methods, such as [3]. In Figure 2, we analysis the relationship between
classification accuracy and class prior. And it shows that our method is robust to the choice of class prior when it is not
far from the true class prior.

[1] Tang Y, You S, Xu C, et al. Bringing Giant Neural Networks Down to Earth with Unlabeled Data. arXiv, 2019.

[2] He X, Cheng J. Learning Compression from Limited Unlabeled Data. ECCV, 2018.

[3] M. C. du Plessis and M. Sugiyama. Class prior estimation from positive and unlabeled data. NIPS, 2014.


[Meta-Review · NeurIPS 2019]

This paper proposed a quite novel application of positive-unlabeled learning: how to compress a big trained teacher network into a small student network, given that the architecture of the teacher network is unknown and the data for distilling the supervision from the teacher network and training the student network is also limited. The proposed method makes use of PU learning by assuming there is a huge pool of data on the cloud regarded as U data; the data uploaded by the user is regarded as P data and then the PU classifier will select more and more related data for distilling the supervision. This only requires the teacher network to be compressed and a limited amount of users' data to be uploaded onto the cloud. As a consequence, this can save a lot of computational resources and may protect some sensitive data from the user side as well. The clarity, the novelty, and the significance are all above the corresponding thresholds of NeurIPS and thus it should clearly be accepted. The problem under consideration is of practical interests and may have huge impacts to our daily life (I assume that many of us are using low-end computing devices everyday). In order to address the broader audience in NeurIPS, many discussions on the motivation should be moved from the rebuttal to the paper itself, including why not compress on the user side and why not directly use the teacher network on the cloud side. The final version should be as friendly as possible to everybody attending the conference. [This meta-review was reviewed and revised by the Program Chairs]